# Sleep in neurointensive care patients, and patients after brain tumor surgery

**Lena Nyholm**[1]*, **Maria Zetterling**[1], **Kristin Elf**[2]

**1** Department of Medical Sciences, Neurosurgery, Uppsala University, Uppsala, Sweden, **2** Department of Medical Sciences, Clinical Neurophysiology, Uppsala University, Uppsala, Sweden

* lena.nyholm@neuro.uu.se

**Data Availability Statement:** The EEG files contain a person-identifying number (person number - a specific individual code identifying each Swedish citizen). Further, the files contain date and time. The person number, and also the information

## Abstract

### Background

Severely brain injured patients treated in the neuro intensive care unit (NICU) are usually sedated. Sedation may affect not only the ability to sleep, but also the EEG rhythms used to identify sleep.

### Aim

The aims were:

1. To study if sleep patterns could be identified in the severely brain injured and sedated patients in the NICU

2. To study if sleep patterns could be identified in patients the night after brain tumor surgery in the neurointermediate care unit (NIMCU)

3. To search for risk factors for not being able to sleep after brain tumor surgery

### Study design

Two populations were included; one with patients affected by severe brain injury and one with patients who had undergone planned brain tumor surgery. This was a quantitative observational study using EEG. Eligible neurointensive care patients for this study had to be suffering from a neurosurgical condition (for example subarachnoid haemorrhage, acute subdural hematoma, intracerebral haemorrhage and meningitis), have affected consciousness and age over 18 years. Thirty-seven patients were included from NICU. Ninety-eight patients, with a suspected glioma (WHO grade II-IV) planned for surgery were also included.

### Results

Neuro intensive care patients, sedated and treated in ventilator, showed no EEG sleep patterns at all. After brain tumor surgery, sleep occurred in 74% of the patients, despite frequent wake-up tests. The patients with sleep patterns were on average 8 years younger, p = 0.03.

about date and time (together with other data given in the article), may each be enough to identify individual study persons. We have so far not been able to convert EEG data into a format without this identifying information. Therefore, publishing the files is currently not compatible with Swedish legislation and ethical approval (https://etikprovningsmyndigheten.se/en/). Data requests may be sent to the head of the Neuro department, Uppsala University Hospital, Uppsala, Sweden (contact via: mats.ryttlefors@akademiska.se).

**Funding:** EEG equipment was funded by The Swedish Medical Society and Olle Engkvist's foundation. The funders had no role in study design, data collection and analysis, decision to publish, or preparation of the manuscript.

**Competing interests:** The authors have declared that no competing interests exist.

## Conclusions

Patients with severe brain injury are at risk of having no sleep when treated at the NICU, whereas after brain tumor surgery, sleep occurs in three-fourths of the patients. Further studies and new methods are warranted to identify sleep and investigate how the loss of sleep affects these patients and how sleep disturbances can be managed.

## Introduction

It is known that critically ill patients have abnormal sleep during intensive care [1]. There are many reasons why patients may suffer from sleep disruption and even sleep deprivation. For example, the neurointensive care environment exposes patients to light, noise, treatment and nursing activities at all hours. Physiological factors such as anxiety, pain, discomfort, loss of physical activity and loss of circadian rhythms make it difficult to sleep [1–3]. In addition, patients in the neurointensive care unit (NICU) are often sedated, which also depresses the ability to sleep [4]. There are many well-known risks associated with sleep deprivation, including delirium [5–7].

Recent studies have found that a proportion of patients treated in the general intensive care unit, even if not sedated, did not have sleep patterns like healthy subjects [8,9]. Sleep occurrence is difficult, or even impossible, to assess in these patients. Attempts have been made to create new sleep classifications to enable sleep detection and sleep evaluation in patients in intensive care, but this has turned out to be difficult [10,11]. Sedatives and other drugs as well as severe illness affect brain activity and the normal sleep patterns and sleep phenomena may not be identifiable [12–14].

An especially vulnerable subgroup of critically ill patients are those with a primary brain disease or injury. In addition to being critically ill and treated in the sleep-hostile intensive care environment, their brain affliction may further interfere with the ability to sleep. To our knowledge, there is no study assessing sleep patterns in these patients, and we wish to draw attention to this field of research with the clinical implication to enable the best possible care and outcome.

### Aim and objectives of study

The aim of this study was to describe sleep patterns monitored with EEG in patients with primary brain disease or injury. Severely brain-injured patients treated in the NICU are usually sedated. Sedation may affect not only the ability to sleep, but also the EEG rhythms used to identify sleep. Therefore, we chose to study two populations, one with sedated patients affected by severe brain injury and one with patients who had undergone planned brain tumor surgery. The latter population have a focal brain disease but they were not sedated postoperatively. Our objectives were:

1. To study if any sleep patterns could be identified in the severely brain-injured and sedated patients in the NICU

2. To study if sleep patterns could be identified in patients the night after brain tumor surgery in the neurointermediate care unit (NIMCU)

3. To search for risk factors for not being able to sleep after brain tumor surgery

## Design and methods

Two populations were included in this study, one with severely brain-injured patients treated at the NICU and one with patients after brain tumor surgery cared for at the NIMCU. The definition of severely brain-injured patients used in this study was that the injury caused unconsciousness and that the patients were sedated and treated in a ventilator. Patients with a disease or injury in the central nervous system who require intensive care and ventilator therapy due to unstable vital parameters are treated at the NICU. All nurses are critical care registered nurses and an anaesthesiologist is always nearby.

In the NICU group, 37 patients were included from 2 November 2015 to 11 April 2016 at the Department of Neurosurgery, Uppsala University Hospital. There were 18 women and 19 men, mean age of 61 (SD 12, range 26–78). All patients were intubated and artificially ventilated. Propofol was given for sedation (maximum 4 mg/kg/h), and morphine hydrochloride for analgesia. Sedation and pain relief were administered until the patient was judged relaxed and comfortable by the nurses, or approximately Richmond Agitation Sedation Scale (RASS) -4 [15]. Level of consciousness and neurological status were recorded three times every 24 hours. On admission to the NICU the mean Glasgow Coma Scale-motor (GCS-M) was 4.8 (SD 0.8, range 2–6) and when EEG monitoring was started mean GCS-M was 4.7 (SD 1.2, range 1–6) [16]. The diagnoses among these patients were subarachnoid haemorrhage (n = 16), acute subdural hematoma (n = 6), intracerebral haemorrhage (n = 5), meningitis (n = 4), intraventricular haemorrhage (n = 2), thromboembolic stroke (n = 2), traumatic subarachnoid haemorrhage (n = 1 and brain tumor (n = 1).

The NICU patients were treated according to a standardized escalated management protocol based on available guidelines [17]. The treatment goals were e.g. intracranial pressure (ICP) < 20 mmHg, cerebral perfusion pressure (CPP) > 60 mmHg and systolic blood pressure (SBP) > 100 mmHg. Reaction levels were checked regularly, ICP was monitored continuously and computed tomography (CT) scans were performed when clinically indicated.

At the NIMCU, post-operative care is provided after neurosurgery. Acutely ill or injured patients who do not require intensive care or ventilator therapy but who need constant supervision by staff are also cared for at NIMCU. The nurses at the unit are registered nurses.

In the NIMCU group, 98 patients were prospectively included. They had a suspected glioma, on preoperative MRI and were planned for surgery at the Department of Neurosurgery, Uppsala University Hospital during the period 22 August 2016 to 7 December 2017. There were 41 women and 57 men with a mean age of 53 (SD 16, range 19–81) years. The surgical procedure is described in previous publications [18,19]. The patients were awakened in the operating theatre and brought to the NIMCU.

The NIMCU patients were monitored by examining the level of consciousness and the presence of neurological deficits every 30 minutes and neurological status according to GCS-M every 60 minutes for the first 6 hours following surgery. Level of consciousness and neurological status were recorded every 60 minutes during hours 6 to 12, and every 120 minutes during hours 12 to 24 after surgery. After 24 hours, the monitoring was prolonged if indicated in selected cases. An acute CT scan was performed in any case of postoperative deterioration or new neurological deficits. In uncomplicated cases, postoperative monitoring continued for 24 hours at the NIMCU and the patient was then discharged to the general ward. If there was a complicated postoperative course, for example with seizures or new neurological deficits, postoperative monitoring in the NIMCU continued until the patient was considered stable [18,19].

Continuous EEG (cEEG) monitoring and simultaneous video recording were collected. In patients treated at the NICU unit, a total of 17 to 21 cup electrodes were pasted and placed

according to the international 10–20 system [20]. In patients undergoing tumor surgery and then monitored in the NIMCU, 9 needle electrodes were inserted subcutaneously under sterile conditions in the operating theatre. All raw cEEGs were reviewed. The cEEG system used was Nicolet One, including Nicolet Monitor in the NICU and Nicolet One Reader for cEEG reading.

## Data analysis

The data analysed in this study had previously been used in two studies [18,21]. In these articles, the occurrence of epilepsy after brain tumor surgery and the occurrence of Stimulus Induced Rhythmic, Periodic, or Ictal Discharges (SIRPIDs) in connection with nursing interventions in NIC patients were presented. For the present study, a secondary analysis was performed and sleep patterns were analysed. A neurophysiologist (K.E.) analysed the first 24 hours of cEEG monitoring for all patients.

Patients treated at the NICU were sometimes disconnected from monitoring due to emergency MRI or other procedures, before which the electrodes had to be removed. Patients with a total of less than 10 hours of EEG monitoring were excluded from this study because shorter than 10 hours may not allow patients to fall asleep. Mean monitored time was 23.2 hours, SD 2.9 hours, Table 1. The standard postoperative care at the NIMCU after brain tumor surgery is 24 hours, although some of these patients had slightly shorter EEG monitoring time (mean 22.4 hours, SD 2.1 hours, Table 2).

The first stage of sleep, non-REM 1 (N1) is characterized by loss of the so-called posterior alpha activity and slowing of background activity in the EEG. The second stage of sleep, non-REM 2 (N2) is typically characterized by sleep phenomena called sleep spindles, Vertex-waves and K-complexes [22]. These sleep phenomena are easy to identify. Since stage N1 may be similar to EEG slowing caused by medications or illness, we used two definitions for sleep. The first included identification of sleep spindles which facilitates sleep identification in patients with pathological EEG slowing. As this actually means identification of N2 sleep, we also used the standard identification of sleep, i.e. stage N1 sleep.

The statistical calculations were done in SPSS. For comparing groups (sex and operated hemisphere) Chi-square test was used with $p < 0.05$ as the level of significance. For comparing age, preoperative tumor volume and grade of resection in percent Mann-Whitney's U-test was used. Differences were considered to be statistically significant when $p < 0.05$.

**Table 1. Demographics and sleep monitoring in neurointensive care (n = 37).**

| | |
|---|---|
| Age in years; Mean (SD, range) | 61 (12, 26–78) |
| Sex (women/men); n (%) | 18 (49) / 19 (51) |
| RASS in all patients | -4 |
| GCS-M on admission to the NICU; Mean (SD, range) | 4.8 (0.8, 2–6) |
| EEG monitored time in hours; Mean (SD) | 23.2 (2.9) |
| Diagnoses: subarachnoid haemorrhage; n (%) acute subdural hematoma; n (%) intracerebral haemorrhage; n (%) meningitis; n (%) intraventricular haemorrhage; n (%) traumatic subarachnoid haemorrhage; n (%) brain tumor; n (%) thromboembolic stroke; n (%) | 16 (43) 6 (16) 5 (13) 4 (11) 2 (5) 1 (3) 1 (3) 2 (5) |

**Table 2. Demographics and sleep monitoring in neurointermediate care (n = 98).**

| | |
|---|---|
| Age in years; Mean (SD, range) | 53 (16, 19–81) |
| Sex (women/men); n (%) | 41 (42) / 57 (58) |
| EEG monitored time in hours; Mean (SD) | 22.4 (2.1) |
| Tumor type and grade; n (%):<br>Glioma WHO grade IV<br>Glioma WHO grade III<br>Glioma WHO grade II<br>Glioma WHO grade I<br>Other (metastasis, lymphoma, unclassified) | <br>49 (50)<br>19 (19)<br>23 (24)<br>1 (1)<br>6 (6) |
| Pre-operative tumor volume ($cm^3$)<br>mean (SD)<br>median ($25^{th}$-$75^{th}$ percentile) | <br>50.0 (52.9)<br>32.2 (10.7–74.9) |
| Tumor location (n)<br>Frontal<br>Temporal<br>Parietal<br>Occipital<br>Insular*<br>Frontal + Corpus callosum/ Gyrus cinguli<br>Frontal-parietal-temporal<br>Temporal-occipital<br>Parietal-temporal<br>Parietal-occipital<br>Midline | <br>33 (34)<br>26 (27)<br>4 (4)<br>2 (2)<br>11 (11)<br>8 (8)<br>1 (1)<br>4 (4)<br>4 (4)<br>4 (4)<br>1 (1) |

*Fronto-insular n = 2, Temp-insular n = 1, Fronto-temporal-insular n = 7, Fronto-temporal-insular + central n = 1.

## Ethics

The local ethics committee approved the study. Included NICU patients gave their informed written consent either themselves or through their next of kin. In the NIMCU group, the responsible surgeon informed about the study the day before surgery and written consent was obtained from the patients at this moment or after some time for consideration, but always before surgery.

## Results

### Sleep patterns in the severely brain-injured and sedated patients at the NICU

None of the monitored patients at the NICU had any sleep patterns or sleep phenomena including both N1 and N2 monitored with EEG. EEG background was generally characterized by slow delta rhythms with periodically superimposed theta or alpha rhythms.

### Sleep patterns in patients after brain tumor surgery at the NIMCU

Sleep duration was not quantified, because a majority of the patients fell asleep and woke up many times during the monitored hours. The range was wide; a few patients merely slept for a few minutes in total while some slept the whole night after surgery with seemingly normal sleep cycles.

### Sleep identified by regular classification (stage N1)

In the patients at the NIMCU, 73 out of 98 patients (74%) had sleep patterns monitored with EEG, Table 3.

**Table 3. Sleep patterns in patients after brain tumor surgery at the NIMCU monitored with EEG (stage N1), N = 98.**

|  | Patients with sleep, n = 73 | Patients without sleep, n = 25 | p-value* |
|---|---|---|---|
| **Age (years)** |  |  |  |
| mean (SD) | 51 (17) | 59 (13) | 0.03 |
| median (range) | 52 (19–81) | 59 (31–81) |  |
| **Pre-operative tumor volume (cm$^3$)** |  |  |  |
| mean (SD) | 49 (54) | 52 (51) | 0.61 |
| median (range) | 29 (0.2–272) | 40 (0.3–238) |  |
| **Grade of resection (%)** |  |  |  |
| mean (SD) | 84 (21) | 81 (24) | 0.44 |
| median (range) | 98 (19–100) | 95 (20–100) |  |

* Mann-Whitney's U-test was used.

## Sleep as identified with sleep spindles (stage N2)

In the patients at the NIMCU, 59 out of 98 patients (60%) had sleep phenomena such as sleep spindles monitored with EEG.

## Risk factors for not being able to sleep after brain tumor surgery

**Sleep identified by regular classification (stage N1).**   The mean age for the patients with sleep patterns was 51 years whereas it was 59 years for the patients without sleep patterns. This difference was significant (p = 0.03), Table 3. The occurrence of sleep patterns did not correlate with sex (p = 0.82), tumor volume (p = 0.61), grade of resection in percent (p = 0.44), or operated hemisphere (p = 0.53).

**Sleep as identified with sleep spindles (stage N2).**   The mean age for the patients with sleep phenomena was 48 years, and for the patients without sleep patterns the mean age was 61 years. This difference was significant (p = 0.001). The occurrence of sleep patterns did not correlate with sex (p = 0.58), preoperative tumor volume (p = 0.59), grade of resection in percent (p = 0.95) or operated hemisphere (p = 0.48).

## Discussion

The main result in this study was that none of the 37 patients with severe brain injury treated at the NICU had any recognizable sleep patterns while 74% of the 98 patients who had undergone planned brain tumor surgery treated at the NIMCU had sleep patterns evident in the EEG. The patients with sleep patterns were significantly younger.

There was no identifiable sleep in the severely brain-injured patients. Their EEGs showed moderated to severe slowing compatible with severely affected brain function and sedation. There is almost no research on how sleep is affected in the acute phase after a brain injury. B M Evans, who compared the REM-system and the arousal changes in the normal and acutely injured brain, concluded that when the NREM-system is damaged the patient becomes unconscious [23].

Synek has provided important understanding of EEG patterns in coma and in different stages of unconsciousness. His most cited work from 1988 describes stages of coma, EEG findings and their prognostic value [13]. Synek's EEG findings may serve as a description of what EEG patterns can be expected in diffuse brain injury. In a subgrade of Synek grade 3, a rare pattern including sleep spindles is included. The sedation, if any, in these patients is not mentioned. However, sleep phenomena in comatose patients have been described.

In accordance with our neurointensive care protocol, all neurointensive care patients were sedated with propofol (maximum 4 mg/kg/h). Propofol is used in intensive care to imitate normal sleep [24], however, propofol reduces deep sleep and REM sleep [25]. Important to this study is that propofol may induce changes in the EEG that resembles Non-REM sleep [12]. This hampers the possibility to identify sleep in propofol-sedated patients.

The main reasons for not identifying any sleep patterns in the EEGs of patients treated at the NICU, may be a result of sedation in addition to their brain injury and coma. Nonetheless, whether these patients slept or not is not possible to say. It has been shown that patients at the general ICU, even if not sedated and being without brain injury, may not show any recognisable sleep when monitored with polysomnography-like methods [8,9]. These results raise important questions about the use of EEG and polysomnographic methods for identifying sleep in severely ill patients. New approaches and better methods to measure and assess sleep in severely ill patients are clearly needed, with the aim of better understanding the consequences of sleep disturbances and what can be done to improve sleep during care in the NICU.

As the severely brain injured patients treated in the NICU did not have any identifiable sleep, it was interesting to study the patients who had undergone brain surgery. The two patient populations have brain disease/injury in common but there was a clear difference in sleep patterns. The most obvious difference is the presence or absence of sedation, but another explanation may be that the post-operative patients have a focal brain injury that originated in a controlled way during the tumor surgery while the NICU patients have focal, multifocal and/or general diffuse brain injuries.

Approximately one fourth of patients operated for brain tumor, did not sleep at all the first day after surgery. In our study, patients who did not fall asleep at the NIMCU were significantly older than those who did fall asleep. In this context, it is important to point out that this was a univariate analysis and it did not consider confounding factors. Studies in healthy people have shown that sleep latency (the time it takes to fall asleep) increases with age, but the increase after the age of 60 is minimal [26]. It is not clear why it was more common among older patients to be awake all night after brain surgery. Perhaps the slightly longer sleep latency increases the probability of being disturbed by noise, light, wake-up tests, pain or other discomfort that can interfere with their falling asleep. This is worrisome because sleep deprivation increases the risk of delirium and may lead to a worse long-term prognosis regarding cognitive outcome [6,8]. Furthermore, there are many other well-known risks associated with sleep deprivation [5–7].

If the brain damage itself makes it impossible for patients with severe brain-injury treated at the NICU to sleep, new questions are raised. When can these patients sleep again? How are they affected by this inability to sleep? What can we do to give these patients the opportunity to sleep as soon as the physiological conditions allow it after the brain injury?

There are studies showing that different types of non-pharmacological nursing interventions can increase sleep among patients cared for in general intensive care [27]. Examples are; minimizing noise during nighttime, providing eye covers and/or ear plugs and promoting daytime wakefulness using for example mobilization and visits by next of kin. The question is whether these interventions could help unconscious patients treated at the NICU. In addition, NICU patients are frequently examined with wake up tests measuring the level of consciousness. This interferes with the ability to fall asleep and stay asleep, while at the same time it is important to detect a deterioration. It is important to balance the patients' opportunity to sleep against the risk of not detecting a neurologic deterioration. This balance is especially important to consider in e.g. elderly patients who have an increased risk of suffering from delirium.

## Limitations

The neurophysiologist was not blinded to the group of patients when reading the EEGs as there was separate data storage for files recorded at the NICU and NIMCU. This is a limitation as blinded reading is preferable to reduce bias.

Patients studied at the NICU were not strictly consecutive e.g. because in some patients, electrodes could not be pasted due to wounds or other causes of increased risk of infection, some patients were to be scanned with MRI (metal cups not compatible), and some patients were expected to be moved to the NIMCU within short. This may have resulted in a selection of patients. Possibly, some sleep patterns might have been detected in a larger sample of patients in which also patients with shallow coma and less sedation had been included.

## Recommendations for practice and further research

Few articles have been published on sleep patterns in NICU patients. The vast majority of studies regarding sleep during intensive care was conducted in patients with an intact brain. Much points to the fact that patients with injuries or diseases in the brain who are in intensive care have different sleep patterns and more research is needed in this regard [23].

## Conclusion

Patients with severe brain injury are at risk of having no sleep when treated at the NICU, whereas after brain tumor surgery, sleep occurs in three-fourths of the patients. Further studies and new methods are warranted to identify sleep and investigate how the loss of sleep affects these patients and how sleep disturbance can be managed.

## Acknowledgments

We thank the staff who assisted us in the data collection, and the technicians and engineers at the unit of clinical neurophysiology.

## Author Contributions

**Conceptualization:** Lena Nyholm, Maria Zetterling, Kristin Elf.

**Data curation:** Lena Nyholm, Maria Zetterling.

**Formal analysis:** Lena Nyholm, Kristin Elf.

**Funding acquisition:** Kristin Elf.

**Methodology:** Lena Nyholm.

**Project administration:** Lena Nyholm.

**Writing – original draft:** Lena Nyholm.

**Writing – review & editing:** Maria Zetterling, Kristin Elf.

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
