## [Decision Letter · Decision Letter 0]

2 Jan 2023

PONE-D-22-30454Sleep in neurointensive care patients, and in postoperative patients after brain tumor surgery

PLOS ONE

Dear Dr. Nyholm,

Thank you for submitting your manuscript to PLOS ONE. After careful consideration, we feel that it has merit but does not fully meet PLOS ONE’s publication criteria as it currently stands. Therefore, we invite you to submit a revised version of the manuscript that addresses the points raised during the review process.

We look forward to receiving your revised manuscript.

Kind regards,

Ayataka Fujimoto

Academic Editor

PLOS ONE

Journal Requirements:

   "EEG equipment was funded by The Swedish Medical Society and Olle Engkvist’s foundation."

   "We thank the staff who assisted us in the data collection, and the technicians and engineers at the unit of clinical neurophysiology. EEG equipment was funded by The Swedish Medical Society and Olle Engkvist’s foundation"

   "EEG equipment was funded by The Swedish Medical Society and Olle Engkvist’s foundation."

Additional Editor Comments:

Please carefully read the comments from the reviewers and revise the manuscript in accordance with the cements as Round 2.

Reviewers' comments:

Reviewer's Responses to Questions

**Comments to the Author**

1. Is the manuscript technically sound, and do the data support the conclusions?

Reviewer #1: Partly

Reviewer #2: Yes

2. Has the statistical analysis been performed appropriately and rigorously? 

Reviewer #1: I Don't Know

Reviewer #2: Yes

3. Have the authors made all data underlying the findings in their manuscript fully available?

Reviewer #1: Yes

Reviewer #2: Yes

4. Is the manuscript presented in an intelligible fashion and written in standard English?

Reviewer #1: No

Reviewer #2: Yes

5. Review Comments to the Author

Reviewer #1: The manuscript “Sleep in neurointensive care patients, and in postoperative patients after brain tumor surgery” is a very interesting work. However, some modifications are desirable to make it better.

1. “postoperative patients after brain tumor surgery” contains the meaning of “after surgery” twice. I think “postoperative” is not needed.

2. The following division is a bit strange: “The diagnoses among these patients were subarachnoid haemorrhage(n=16), acute subdural hematoma (n=6), intracerebral haemorrhage (n=4), meningitis (n=4), intraventricular haemorrhage (n=2), cerebellar bleeding (n=1), traumatic subarachnoid haemorrhage (n=1), brain tumour (n=1), stroke (n=1) and cerebellar infarction (n=1)” in the Design and methods. Cerebellar bleeding usually is included intracerebral haemorrhage. Stroke shows cerebral haemorrhage or infarction or subarachnoid haemorrhage. What does stroke (n=1) mean? Please correct it.

3. “severely brain-injured” is a keyword in this article. However, it is unclear what “severely brain-injured” means. You should describe the definition of “severely brain-injured” in the Method.

4. It is better to explain the difference between NICU and NIMCU.

5. You mentioned that patients in the NIMCU were prospectively included. If this article is a prospective study or a study using prospectively registered data, you need to describe about obtaining written informed consent in the Ethics.

6. Please show references about the sleep stage (N1, N2).

7. You mentioned that ”On admission to the NICU the mean Glasgow Coma Scale motor (GCS-M) was 4.8 (range 2-6) and when EEG monitoring was started mean GCS was 4.7 (range 1-6)(16).” Why did you use GCS-motor, instead of GCS? I think using GCS makes it easier to understand the degree of impaired consciousness. You should use GCS or explain the reason to use GCS-motor. And, in “mean GCS was 4.7 (range 1-6),” the minimum of GCS is 3. Is range 1-6 correct? (or do you mean GCS-motor?)

8. I think one of the important factors that affect sleep just after surgery is pain. Please consider describing VAS or NRS and analyzing if possible.

9. Please add more information to the main results. For example, “tumor volume (p=0.61)” can be modified like “tumor volume (XX±XX cm3 in patients with sleep vs YY±YY cm3 in patients without sleep, p=0.61).” Moreover, for emphasis, it would be better to present the main results in a Table.

10. How about changing “Monitored time in hours” in Table 1 to “EEG Monitored time in hours”? I think this is more understandable.

11. The significance of the following result is not clear because all patients in the NICU group were intubated and controlled at RASS -4: “None of the patients in the NICU had any sleep patterns or sleep phenomena monitored with EEG.” It seems to go without saying that EEG can’t identify the sleep stage under deep sedation or severe disturbance of consciousness. It is required to describe persuasive content in the Discussion while citing past literature.

12. You mentioned that “It is important to balance the patients' opportunity to sleep against the risk of not detecting a neurologic deterioration.” I know intuitively that these patients also need sleep. However, it would be better to cite references showing that it is important to get some sleep even in patients requiring neuromonitoring.

13. There are some grammatical mistakes as blow. Please double-check your manuscript.

・The sentence in Aim and objectives of study “Sedation may affect not the ability to sleep, but also the EEG rhythms used to identify sleep” should be modified to “Sedation may affect not ONLY the ability to sleep, but also the EEG rhythms used to identify sleep.”

・In this sentence “All patients who were not responding to commands, Glasgow Coma scale - Motor (GCS-M)(16) <5, were intubated, sedated and artificially ventilated”, I think “who” is not needed.

Reviewer #2: Summary of article:

Dr. Nyholm et al. have investigated the sleep pattern in patients with severe brain injury and sedated and in patients who underwent elective brain tumor resection surgery. Using two separate datasets of patients treated in the neuro intensive care unit and patients who underwent elective brain tumor resection surgery, the authors addressed three following questions: (1) whether and how amount the patients with severe brain injury and sedated have normal patterns of sleep; (2) whether and how amount the patients with the resection surgery regional brain tumor and postoperative non-sedation have normal patterns of sleep; (3) whether there is any risk factor for not being able to sleep. As a result, the authors found that (1) none of the 37 patients in the neurointensive care unit showed normal sleep patterns; (2) 59/73 of 98 patients with the resection surgery regional brain tumor and postoperative non-sedation showed normal stage N1/ stage N2 sleep patterns, respectively; (3) Among patients with the resection surgery regional brain tumor and postoperative non-sedation, patients with normal sleep patterns were younger than without them. The authors concluded that older patients were more likely not to fall asleep at all.

Comments (Invitation on Dec 27, 2022, and comment submission on Dec 28, 2022)

This study addressed an interesting topic for sleep patterns in patients with brain injury, tumor, or sedation. However, there are some concerns for publication. Therefore, please consider addressing some concerns, as shown below.

Here are my comments and suggestions about this manuscript.

Major points:

[1] “Introduction”

This study consists of three aims. Therefore, it is better to describe the purpose/aim, methods, and results of each experiment in the manuscript, such as the followings:

[Aim 1]

[Aim 2]

[Aim 3]

[2] “Abstract”

Please rewrite the abstract so readers can understand this study's essence only by reading the abstract. The authors might want to refer to the following advice;

(1) It is better to describe these three aims in the abstract.

(2) It is better to clarify this study’s outcome.

(3) As for the explanation of two datasets, it is better to state in [Study design] rather than [Background].

(4) It is better to show specific numbers and statistic values in [Results].

(5) Please avoid redundant expressions between [Results] and [Conclusion].

[3] “Introduction”

Please clarify the clinical relevance or implication of this study in the Introduction.

[4] “Methods”

Please describe why the inclusion periods differed for patients in the NICU and NIMCU groups. In addition, please identify the name of the hospital of the NICU group.

[5] “Methods”

Please clarify whether a neurophysiologist was blind to the group of patients reading cEEG monitoring. If not blind, please add that point as a limitation.

[6] “Table 1”

Please separate the tables for the group of the neuro intensive care unit and neuro intermediate care unit. This is because these datasets were used for different aims. In addition, please add the rows of the RASS scale and cause of admission in the table for the neuro intensive care unit group and rows of tumor type, grade, volume, region of brains, past medical history of insomnia, preoperative daily use of medications for insomnia in the table for the group of the neuro intermediate care unit.

[7] “Limitation”

Please add the limitation that the statistical significance of the mean age difference between patients with and without sleep was univariate analysis. This analysis did not consider confounding factors.

Minor points:

[8] “Methods”

It is helpful for broad readers to briefly explain how the neuro intensive care unit and neuro intermediate care unit are defined or the inclusion criteria for each ward.

[9] “typo”

Abstract: not no fall asleep -> not fall asleep

6. PLOS authors have the option to publish the peer review history of their article (what does this mean?). If published, this will include your full peer review and any attached files.

Reviewer #1: No

Reviewer #2: **Yes: **Naoto Kuroda

---

## [Author Response · Author response to Decision Letter 0]

12 Feb 2023

Response: We have now adapted to the style requirements

Response: Details regarding participant consent is inserted in the ethics statement in the Methods and in the online submission information

 "EEG equipment was funded by The Swedish Medical Society and Olle Engkvist’s foundation."

Response: This statement is inserted in the cover letter: “The funders had no role in study design, data collection and analysis, decision to publish, or preparation of the manuscript.”

 "We thank the staff who assisted us in the data collection, and the technicians and engineers at the unit of clinical neurophysiology. EEG equipment was funded by The Swedish Medical Society and Olle Engkvist’s foundation"

 "EEG equipment was funded by The Swedish Medical Society and Olle Engkvist’s foundation."

Response: Thank you. The funding information is removed from the manuscript and placed in the cover letter. 

Additional Editor Comments:

Please carefully read the comments from the reviewers and revise the manuscript in accordance with the cements as Round 2.

Response: Data availability: The EEG files contain a person-identifying number (person number - a specific individual code identifying each Swedish citizen). Further, the files contain date and time. The person number, and also the information about date and time (together with other data given in the article), may each be enough to identify individual study persons. We have so far not been able to convert EEG data into a format without this identifying information. Therefore, publishing the files is currently not compatible with Swedish legislation and ethical approval (https://etikprovningsmyndigheten.se/en/). Data requests may be sent to the head of the Neuro department, Uppsala University Hospital, Uppsala, Sweden.

1. “postoperative patients after brain tumor surgery” contains the meaning of “after surgery” twice. I think “postoperative” is not needed.

Response: Thank you for bringing this to our attention. It is now corrected.

2. The following division is a bit strange: “The diagnoses among these patients were subarachnoid haemorrhage(n=16), acute subdural hematoma (n=6), intracerebral haemorrhage (n=4), meningitis (n=4), intraventricular haemorrhage (n=2), cerebellar bleeding (n=1), traumatic subarachnoid haemorrhage (n=1), brain tumour (n=1), stroke (n=1) and cerebellar infarction (n=1)” in the Design and methods. Cerebellar bleeding usually is included intracerebral haemorrhage. Stroke shows cerebral haemorrhage or infarction or subarachnoid haemorrhage. What does stroke (n=1) mean? Please correct it.

Response: Cerebellar bleeding is now included intracerebral haemorrhage and STROKE is clarified with thrombectomy.

3. “severely brain-injured” is a keyword in this article. However, it is unclear what “severely brain-injured” means. You should describe the definition of “severely brain-injured” in the Method.

Response: Severely brain-injured patients is now defined as patients who were sedated and treated in a ventilator due to their brain damage. This is clarified in the design and methods. 

4. It is better to explain the difference between NICU and NIMCU.

Response: This is now explained more clearly in the design and methods

5. You mentioned that patients in the NIMCU were prospectively included. If this article is a prospective study or a study using prospectively registered data, you need to describe about obtaining written informed consent in the Ethics.

Response: This information is now included in the Ethics.

6. Please show references about the sleep stage (N1, N2).

Response: The main reference explaining the scientific ground for the sleep stages definitions is now provided.

7. You mentioned that ”On admission to the NICU the mean Glasgow Coma Scale motor (GCS-M) was 4.8 (range 2-6) and when EEG monitoring was started mean GCS was 4.7 (range 1-6)(16).” Why did you use GCS-motor, instead of GCS? I think using GCS makes it easier to understand the degree of impaired consciousness. You should use GCS or explain the reason to use GCS-motor. And, in “mean GCS was 4.7 (range 1-6),” the minimum of GCS is 3. Is range 1-6 correct? (or do you mean GCS-motor?)

Response: We presented GCS-M on the basis that the department where the study was conducted routinely uses the Reaction level scale (RLS). RLS is directly translatable to GCS-M. These patients were all intubated, which made verbal response not possible. Many of these patients are also unable to open their eyes due to swelling. We have data for a complete GCS on arrival but not on EEG start. With all this taken together, we choose to continue reporting GCS-M.

Ref. Starmark JE, Stålhammar D, Holmgren E. The Reaction Level Scale (RLS85). Manual and guidelines. Acta Neurochir (Wien). 1988;91(1-2):12-20. doi:10.1007/BF01400521

8. I think one of the important factors that affect sleep just after surgery is pain. Please consider describing VAS or NRS and analyzing if possible.

Response: Unfortunately, we do not have this data available. However, all patients were offered medications, i.e., morphine to control the postoperative pain.

9. Please add more information to the main results. For example, “tumor volume (p=0.61)” can be modified like “tumor volume (XX±XX cm3 in patients with sleep vs YY±YY cm3 in patients without sleep, p=0.61).” Moreover, for emphasis, it would be better to present the main results in a Table.

Response: We have now added this information, please see table 3.

10. How about changing “Monitored time in hours” in Table 1 to “EEG Monitored time in hours”? I think this is more understandable.

Response: This is corrected.

11. The significance of the following result is not clear because all patients in the NICU group were intubated and controlled at RASS -4: “None of the patients in the NICU had any sleep patterns or sleep phenomena monitored with EEG.” It seems to go without saying that EEG can’t identify the sleep stage under deep sedation or severe disturbance of consciousness. It is required to describe persuasive content in the Discussion while citing past literature.

Response: The request to discuss and provide literature references regarding the ability or disability to identify the sleep stages under deep sedation and/or severe disturbance of consciousness is acknowledged and appreciated. New paragraphs pertaining to these issues have been added to the discussion. Some of the text regarding the limitations of EEG to detect sleep in comatose brain injured patients, formerly placed under Limitations, have been moved up to the general discussion as this rather belongs to the requested discussion, than constitute a limitation of the study performance. Further, a new paragraph has been added to Limitations, as we have identified a source of selection bias that might have contributed to the results (absence of sleep phenomena in NICU patients).

12. You mentioned that “It is important to balance the patients' opportunity to sleep against the risk of not detecting a neurologic deterioration.” I know intuitively that these patients also need sleep. However, it would be better to cite references showing that it is important to get some sleep even in patients requiring neuromonitoring.

Response: Thank you for pointing this out, we have now elaborated on this and included relevant references.

13. There are some grammatical mistakes as blow. Please double-check your manuscript.

・The sentence in Aim and objectives of study “Sedation may affect not the ability to sleep, but also the EEG rhythms used to identify sleep” should be modified to “Sedation may affect not ONLY the ability to sleep, but also the EEG rhythms used to identify sleep.”

・In this sentence “All patients who were not responding to commands, Glasgow Coma scale - Motor (GCS-M)(<5, were intubated, sedated and artificially ventilated”, I think “who” is not needed.

Response: This is now corrected.

[1] “Introduction”

This study consists of three aims. Therefore, it is better to describe the purpose/aim, methods, and results of each experiment in the manuscript, such as the followings:

[Aim 1]

[Aim 2]

[Aim 3]

Response: Thank you for this suggestion. The aim is now divided into three objectives and the rest of the text is corrected to keep that organization as best as possible.

[2] “Abstract”

Please rewrite the abstract so readers can understand this study's essence only by reading the abstract. The authors might want to refer to the following advice;

(1) It is better to describe these three aims in the abstract.

(2) It is better to clarify this study’s outcome.

(3) As for the explanation of two datasets, it is better to state in [Study design] rather than [Background].

(4) It is better to show specific numbers and statistic values in [Results].

(5) Please avoid redundant expressions between [Results] and [Conclusion].

Response: Thank you for this advice, we have now implemented most of it.

[3] “Introduction”

Please clarify the clinical relevance or implication of this study in the Introduction.

Response: We have now clarified the clinical relevance at the end of the Introduction.

[4] “Methods”

Please describe why the inclusion periods differed for patients in the NICU and NIMCU groups. In addition, please identify the name of the hospital of the NICU group.

Response: This information is now added.

[5] “Methods”

Please clarify whether a neurophysiologist was blind to the group of patients reading cEEG monitoring. If not blind, please add that point as a limitation.

Response: Information that the EEG reader was not blinded to group has been added to methods and is also problematized under Discussion, Limitations. 

Please separate the tables for the group of the neuro intensive care unit and neuro intermediate care unit. This is because these datasets were used for different aims. In addition, please add the rows of the RASS scale and cause of admission in the table for the neuro intensive care unit group and rows of tumor type, grade, volume, region of brains, past medical history of insomnia, preoperative daily use of medications for insomnia in the table for the group of the neuro intermediate care unit.

Response: This information is added except information of past medical history of insomnia, preoperative daily use of medications for insomnia in the table for the group of the neuro intermediate care unit. This is because we unfortunately do not have access to that information

Please add the limitation that the statistical significance of the mean age difference between patients with and without sleep was univariate analysis. This analysis did not consider confounding factors.

Response: This limitation is added in the Discussion. 

[8] “Methods”

It is helpful for broad readers to briefly explain how the neuro intensive care unit and neuro intermediate care unit are defined or the inclusion criteria for each ward.

Response: This is now explained more clearly in the design and methods

[9] “typo”

Abstract: not no fall asleep -> not fall asleep

Response: This is corrected

---

## [Decision Letter · Decision Letter 1]

16 May 2023

Sleep in neurointensive care patients, and patients after brain tumor surgery

PONE-D-22-30454R1

Dear Dr. Nyholm,

We’re pleased to inform you that your manuscript has been judged scientifically suitable for publication and will be formally accepted for publication once it meets all outstanding technical requirements.

Kind regards,

Ayataka Fujimoto

Academic Editor

PLOS ONE

Additional Editor Comments (optional):

Since Reviewer#3 stated "reject" but #1 and #2 have accepted,  the author later responded to the comments from Reviewer#3 that was reasonable. Therefore, my final decision is "accept".

Reviewers' comments:

Reviewer's Responses to Questions

**Comments to the Author**

1. If the authors have adequately addressed your comments raised in a previous round of review and you feel that this manuscript is now acceptable for publication, you may indicate that here to bypass the “Comments to the Author” section, enter your conflict of interest statement in the “Confidential to Editor” section, and submit your "Accept" recommendation.

Reviewer #1: All comments have been addressed

Reviewer #3: (No Response)

2. Is the manuscript technically sound, and do the data support the conclusions?

Reviewer #1: Yes

Reviewer #3: No

3. Has the statistical analysis been performed appropriately and rigorously? 

Reviewer #1: Yes

Reviewer #3: No

4. Have the authors made all data underlying the findings in their manuscript fully available?

Reviewer #1: Yes

Reviewer #3: No

5. Is the manuscript presented in an intelligible fashion and written in standard English?

Reviewer #1: No

Reviewer #3: Yes

6. Review Comments to the Author

Reviewer #1: Thank you for your revision. Please modify it a little more.

1. “1. Aim: The aims were: To study if sleep patterns could be identified in the severely

brain injured and sedated patients in the NICU” in the Abstract seems incorrect sentence. Please modify this sentence as below, if it is correct.

“Aim: The aims were:

1. To study if sleep patterns could be identified in the severely brain injured and sedated

patients in the NICU”

2. Regarding the sentence in the 5th paragraph of Design and methods, “In uncomplicated cases, postoperative monitoring continued for 24 hours in the NICU and the patient was then discharged to the general ward”, I wondered if “the NICU” is a mistake of “the NMICU”. If it is a mistake, please correct it.

3. There are unnecessary underlines as below. Please correct.

・”Study design: Two populations were included; one with patients affected by severe brain injury and one with patients who have undergone planned brain tumour surgery._This was a …” in the Abstract

・“Perhaps the slightly longer sleep latency increases the probability of being disturbed by noise, light, wakeup tests, pain or other discomfort that can interfere with them falling asleep._This is worrisome…” in the Discussion.

4. There are still many careless mistakes, although I advised you to double-check. Please modify.

・Please change the sentence “This was a uantitative observational study with EEG.” to “This was a quantitative observational study with EEG” in the Abstract.

・Missing parentheses: “traumatic subarachnoid haemorrhage (n=1” in the Design and method.

・Comma should be changed to a period in this sentence “Propofol is used in intensive care to imitate normal sleep (24), However, propofol reduces deep sleep and REM sleep (25).” “Propofol is used in intensive care to imitate normal sleep (24). However, propofol reduces deep sleep and REM sleep (25)” seems correct.

・Missing period: “This is worrisome because sleep deprivation increases the risk of delirium and may lead to a worse long term prognosis regarding cognitive outcome (6,8)”

The statistical analysis in Table 3 is a little confusing. T-test appeared to be performed because p-value is described in the line of mean (SD) row. However, you mentioned you used Mann-Whitney U-test. Generally, Mann-Whitney U-test is utilized for the median (IQR) and the t-test for the mean (SD). It would be better to explain which analysis you used with footnotes or to describe the p-value in the line of the median row.

Reviewer #3: Comments (invitation: April 5, 2023, and submission: April 9. 2023)

The authors investigated the sleep patterns of sedated severe brain injury patients and non-sedated postoperative brain tumor patients. However, as noted below, the study's background and objectives are not entirely consistent, and it is unclear whether the study's aims were achieved through the methods employed. Further modifications are discussed below.

1. Abstract: One of the aims of this study was to adequately explain the sleep patterns in patients with brain injuries. As stated by the authors themselves in the limitations and "Recommendations for practice and further research", the impact of sedation may have affected the results. Therefore, the authors' findings did not achieve the research objective.

2. Abstract, Introduction, and Aim: The background and objectives of the study do not align. If the authors aim to investigate the impact of sedation as described in the background, a comparison of sedated brain injury patients with sedated brain tumor patients would be appropriate. Alternatively, if their objective is to study sleep patterns, a more suitable approach would be to examine non-sedated brain injury patients and non-sedated postoperative brain tumor patients.

3. Methods and Results: As the results for sedated patients are not well-described, it might be more straightforward to focus solely on the postoperative group of brain tumor patients. However, if the authors intend to include sedated patients in their study, a more detailed description of their results would be necessary to ensure clarity and understanding.

4. Methods and Results: Although the study reports a correlation between age and postoperative sleep patterns, it would be beneficial to include the statistical values used in the analysis, in addition to the p-values, in the Table for greater clarity.

5. Methods and Results: In the present study, it would be more appropriate to conduct a logistic regression analysis with the occurrence of sleep patterns as the dependent variable, and age, tumor volume, and resection rate as independent variables. Furthermore, a multivariate analysis would be beneficial to account for the potential influence of additional factors, such as anesthesia duration, in addition to the current variables.

6. Methods and Results: Is analyzing N1 and N2 sleep stages adequate to determine sleep patterns? If so, please provide relevant literature demonstrating this.

7. Limitations: As the authors have acknowledged, it is unclear if sedated severe brain injury patients actually slept during the study. Given that this issue is central to the study's aim, as stated in the abstract, the authors should either compare sedated brain injury patients with sedated brain tumor patients or investigate non-sedated brain injury patients and non-sedated postoperative brain tumor patients to address this uncertainty.

8. Conclusion: It is difficult to determine whether a severe brain injury affects sleep patterns without comparing it to a sedated individual without brain injury.

9. Figure: It would be beneficial to include EEG figures for both sedated patients and post-operative brain tumor patients. Additionally, it would be helpful to provide EEG figures for both younger and older patients in the post-operative brain tumor group.

7. PLOS authors have the option to publish the peer review history of their article (what does this mean?). If published, this will include your full peer review and any attached files.

Reviewer #1: No

Reviewer #3: No

---

## [Editor Report · Acceptance letter]

7 Jun 2023

PONE-D-22-30454R1 

Sleep in neurointensive care patients, and patients after brain tumor surgery 

Dear Dr. Nyholm:

I'm pleased to inform you that your manuscript has been deemed suitable for publication in PLOS ONE. Congratulations! Your manuscript is now with our production department. 

Kind regards, 

on behalf of

Dr. Ayataka Fujimoto 

Academic Editor

PLOS ONE